# Music Therapy in Global Aphasia: A Case Report

**DOI:** 10.3390/medicines10020016

**Published:** 2023-01-23

**Authors:** Adriana Piccolo, Francesco Corallo, Davide Cardile, Michele Torrisi, Chiara Smorto, Simona Cammaroto, Viviana Lo Buono

**Affiliations:** IRCCS Centro Neurolesi “Bonino-Pulejo” S.S. 113 Via Palermo, C/da Casazza, 98124 Messina, Italy

**Keywords:** global aphasia, music therapy, rehabilitation

## Abstract

Patients affected by global aphasia are no longer able to understand, produce, name objects, write and read. It occurs as a result of functional damage of ischemic or hemorrhagic origin affecting the entire peri-silvan region and frontal operculum. Rehabilitation training aims to promote an early intervention in the acute phase. We described a case of a 57-year-old female patient with left intraparenchymal fronto-temporo-parietal cerebral hemorrhage and right hemiplegia. After admission to clinical rehabilitative center, the patient was not able to perform simple orders and she presented a severe impairment of auditory and written comprehension. Eloquence was characterized by stereotypical emission of monosyllabic sounds and showed compromised praxis-constructive abilities. Rehabilitation included a program of Neurologic Music Therapy (NMT), specifically Symbolic Communication Training Through Music (SYCOM) and Musical Speech Stimulation (MUSTIM). Rehabilitative treatment was measured by improved cognitive and language performance of the patient from T0 to T1. Music rehabilitative interventions and continuous speech therapy improve visual attention and communicative intentionality. In order to confirm the effectiveness of data presented, further extensive studies of the sample would be necessary, to assess the real role of music therapy in post-stroke global aphasia.

## 1. Introduction

Global aphasia is a severe disorder involving profound impairment of all modes of receptive and expressive language [1]. It occurs as a result of functional damage of ischemic or hemorrhagic origin affecting the entire peri-silvan region and frontal operculum [2]. Patients affected by global aphasia typically have marked impairment in comprehension of single words, sentences and conversations and severely limited oral production, writing and reading [3,4].

Although this form of language impairment can evolve into an aphasic syndrome of lesser severity [5], the disorder still remains highly disabling for the patient, as it worsens quality of life impairing social skills, rehabilitative outcomes and socio-occupational reintegration.

A rehabilitation training in the acute phase appears to correlate with better outcomes [6]. Speech and language therapy results in statistically significant improvements in communication, reading, writing and expressive language [7].

Music and music therapy may improve communication skills in different neurological disorders [8,9]. The inclusion of sound and music within the rehabilitation process is associated with the capacity to stimulate brain areas involved in emotional processing and motor control, such as the fronto-parietal network [10]. In aphasia, melody-based treatments rely on the notion that they can induce structural white matter neuroplasticity and that right hemisphere (with preserved musical abilities) can have compensatory role. Bitan et al. [11] show how the use of melody-based treatment results in increased connectivity between motor speech control areas (bilateral supplementary motor areas and insulae) and right hemisphere language areas (inferior frontal gyrus pars triangularis and pars opercularis). Sihvonen et al. [12] found that a 3-month neurological music therapy treatment resulted in increased quantitative anisotropy (QA) in the right dorsal pathways (arcuate fasciculus and superior longitudinal fasciculus) and in the corpus callosum and the right frontal aslant tract, thalamic radiation and corticostriatal tracts.

Music rehabilitative interventions range from the use of rhythmic auditory stimulation to facilitate the movement and normalization of gait parameters [13], to music listening and singing to reduce pain and improve sense of well-being [14,15].

In non-fluent aphasia, numerous studies have also confirmed the effectiveness of music therapy intervention, particularly with regard to the improvement of joints, prosody, respiratory and vocal capacity [16,17]. Moreover, in these patients, the treatment generates an improvement in cognition and motor function, alleviate negative moods and accelerate neurological recovery [18].

Concerning global aphasia, no treatment has been shown to be effective in language recovery.

The literature shows a strong correlation between language and subcortical lesions, especially in the basal ganglia, which are involved in rhythm processing, temporal prediction, motor programming and execution [19]. For this reason, in this study, we described the effects of an individualized music therapy treatment in a patient with total suppression of language and absence of non-verbal communication after a cerebrovascular event. The resource-orientated and training-centered combination treatment was based on parallel processing strategies between music and language. We aimed to verify whether music rehabilitation could create a positive effect to improve cognitive and language performance.

## 2. Case Description

A 57-year-old female was admitted to our clinical rehabilitative center with right hemiplegia following a stroke.

She was not able to perform simple orders and presented a severe impairment of auditory and written comprehension. Eloquence was characterized by stereotypical emission of monosyllabic sounds and showed compromised praxis-constructive abilities. Concerning facial play, gesture or vocal utterances were absent. The gestures were mostly aimless, with a bodily attitude of closure. Psychological findings showed a depressive mood with frequent episodes of crying, oppositive behavioral and apathy.

The MRI that the patient underwent showed an extensive foci with inhomogeneous signal in left capsular nucleus region and ipsilateral insular fronto-parietal region, with cortico-subcortical distribution, predominantly hyper intense in long TR scans with more hypo intense components in T2 referable to outcome of previous ischemic event with hemorrhagic infarction in patient with suspected cavernomatosis. The MRI image is reported in Figure 1.

Patient was evaluated before (T0) and after (T1) the 6-month rehabilitative program. The assessment was conducted by administering the following tests:

Token test [20], Aphasic depression rating scale (ADRS) [21], Levels of Cognitive Functioning (LCF) [22] and Coma Recovery Rating Scale revised (CRS-R) [23]. We also used a communication scale proposal to monitor non-verbal communication. This scale measured the production of nonverbal communicative acts (e.g., gestures, directions, responses through head or eye movements) and positive or negative emotional manifestations (smiling, crying, glances, annoyance, etc.) and recording their frequency of occurrence.

The rehabilitative program consisted of three parts: cognitive rehabilitation, speech therapy and neurological music therapy (NMT). It was administered by a psychologist and a speech-therapist for a period of 6 months, two times per week.

The cognitive rehabilitative session was focused on attention through ATP (attention process training). The latter is a multi-session exercise designed to improve the speed of processing information and the ability to focus on relevant material while ignoring distractions. In this case, patient listening an audio track presenting a variety of stimuli. Subsequently he was asked to press a buzzer when the target stimulus was given, at a time interval that was decreasing with time.

The structured speech therapy rehabilitation was focused on phonemic discrimination exercises, naming and recognition of objects and figures, in line with what was reported by Beaulieu et al. [24].

The Neurologic Music Therapy (NMT) consists of a therapeutic application of music stimulation through standardized techniques to bring improvements in several cognitive domains. One of the main goal areas of this approach is speech and language treatment, with several techniques (Table 1).

Among these techniques, because of the patient’s pathology was decided to use SYCOM and MUSTIM. During Symbolic Communication Training Through Music (SYCOM) the patient was asked to play a certain pattern on their instrument to communicate a certain word or phrase of increasing difficult (e.g., “Yes”: hit once, “No”: Hit twice, “Maybe”: hit three times). After an initial training phase on these associations, the patient was asked to answer simple questions using the learned patterns. This non-verbal “language” system simulates and train appropriate communication behaviors, language pragmatics, speech gestures and emotional expression.

These exercises can be used to train structural communication behavior, such as dialoguing, asking questions and creating answers, listening and responding, appropriate speech gestures and other communication structures in social interaction patterns in real time. For the Musical Speech Stimulation (MUSTIM), to make the treatment tailored to the patient we collected all the information related to the patient’s personal history and his favorite type of music. During these sessions, patient’s favorite songs (or other famous songs) was used, then stopped at a certain point in order to stimulate the desire to fill in the blanks. This is a very common technique for non-fluent aphasia because it triggers automatic speech and stimulate long term memory.

The effect of rehabilitation treatment was measured by improved cognitive and language performance of the patient between the first (T0) and last (T1) assessment.

Music rehabilitative interventions and continuous speech therapy improve visual attention and communicative intentionality.

The score obtained on the ADRS scale improves (from 13 to 10), but still remains above the cut-off (x < 9/32) (see Table 2). On the SCNV there is an improvement from 19 (0< x < 21 = low) to 27 (22 < x < 42 moderate) resulting in increased communication skills, being able most of the time to maintain posture, eye contact and use facial expressions. Specifically, social gestures (such as greetings) had become part of the patient’s daily behavioral repertoire. The increase obtained at the score for the LCF scale (from 5 to 6) shows an improvement at the level of cognitive retrieval, which remains fuzzy, but goes from being inappropriate (LCF = 5) to appropriate (LCF = 6). There is also an improvement on the token test, whose score goes from 3 to 4, but still remains strongly below the threshold (cut-off = 29).

## 3. Discussion

Traditional neurorehabilitation approach aims to functional recovery in various forms and severity of aphasia; however, these cognitive and speech therapy interventions, in cases of global aphasia, may not always show the desired benefits [25].

Non-verbal communication allows to directly grasp the intentions and desires of others, without the use of spoken language. The areas implicated in the understanding of non-verbal language are upper and left medial temporal lobe together with the thalamus and the fronto-parietal mirror neuron system’ (premotor cortex and left lower parietal cortex) [26]. Music therapy seems to act on these areas through facial gestures and tonal/rhythmic inflection [16].

Furthermore, the emotional component of music activates different brain areas associated with emotional behaviors, such as insular and cingulate cortex, hypothalamus, hippocampus, amygdala and prefrontal cortex [2,25].

Several studies support the use of singing and intoning to trigger non-propositional speech in people who present a non-fluent aphasia. Liu et al. [27] in a recent review and meta-analysis found a significant effect in functional communication, repetition and in naming, but not in comprehension.

Other studies [28,29] found that singing helped word phrase production in same patients with severe expressive aphasia, probably due to the association of melody and text in long-term memory. In a case study conducted by Yamaguchi et al. [30], the results indicated that singing can be an effective treatment for severe non-fluent aphasia in rehabilitative therapy in a patient with cognitive impairment.

In this study, the patient did not show verbal recovery or the ability to repeat vocal material for known songs. On the contrary, we observed a marked improvement in nonverbal communication skills and expressive emotional range as evidenced by increased use of gestures and facial expressions and more consistent interaction with the carer.

The motivation for the need to communicate had extended from the primary physiological needs to the expression of the patient’s desires. Achieving results during the rehabilitation sessions, made the patient more confident about their abilities. This could result in a benefit of sense of self-efficacy and an improvement in mood tone and in relationships with friends and caregivers and in socialization. It is likely to assume that this result is attributable to the specific effect of music therapy. Indeed, cognitive and speech therapy focus more on merely verbal skills.

Therefore, an individualized approach in post-stroke rehabilitation, such as music therapy, may be particularly useful. Despite there being a relatively small portion of stroke patients with poststroke aphasia, it has been stated that global aphasia is more isolating and debilitating than blindness or hemiplegia cases of global aphasia. In fact, language skills include not only the production of words and phrases, but also interaction with the eyes, facial expression, gestures, proxemics and posture, which could be an aid to stimulate alternative communication strategies, especially in cases where no verbal recovery occurs. In particular, proxemics and gestural communication could be effective in understanding simple and complex orders, thus improving therapeutic compliance and mood in patients with global aphasia.

In order to confirm the effectiveness of data presented, further and more extensive studies of the sample would be necessary to assess the real role of music therapy in post-stroke global aphasia. For example, one could investigate on a larger sample the effects that different NMT techniques produce at the level of brain plasticity. Another interesting aspect might be to investigate the effects on different clinical populations.

## Figures and Tables

**Figure 1 medicines-10-00016-f001:**
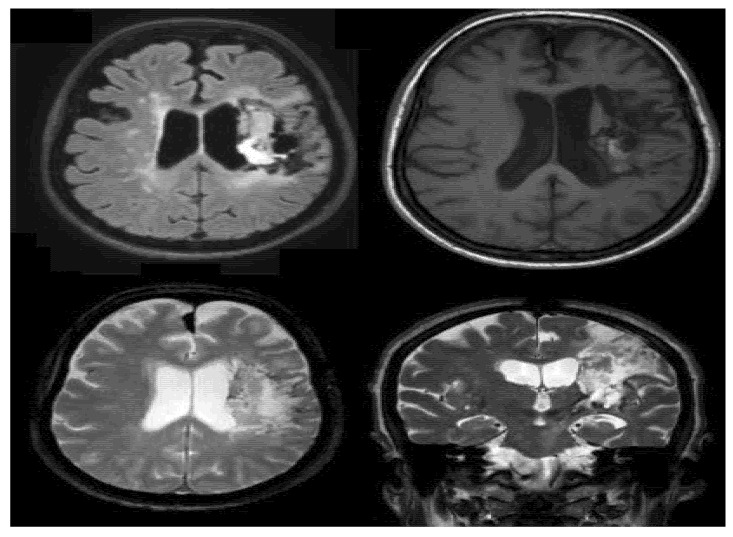
Patient’s brain damage evidenced by MRI scan.

**Table 1 medicines-10-00016-t001:** NMT speech and language techniques.

NMT Techniques	Description
Developmental Speech and Language Training through Music (DSLM)	The use of musical materials and experiences to enhance speech and language development. Technique appropriate specially for children with autism and disabilities.
Melodic Intonation Therapy (MIT)	A three-level approach to improve fluent output of language for clients with severe Broca’s aphasia.
Musical Speech Stimulation (MUSTIM)	Utilization of musical materials (such as songs, rhymes and chants) to stimulate non propositional speech.
Oral Motor and Respiratory Exercises (OMREX)	Sound vocalization exercises and wind instrument playing to strength and coordination in making speech sounds. Technique appropriate for patients with apraxia, cerebral palsy, and respiratory problems.
Rhythmic Speech Cueing (RSC)	Utilization of metric/patterned rhythmic cues to control speech rate and to facilitate initiation of speech. Appropriate for clients with apraxia, dysarthria and fluency disorders.
Symbolic Communication Training through Music (SYCOM)	Use of structured experiences in instrumental or vocal improvisation to train communication behavior. Technique appropriate for patients who may not develop speech but could still master or re-gain language concepts.
Therapeutic Singing (TS)	Singing activities to practice speech articulation and improve respiratory function. This technique is appropriate for patients with apraxia, dysarthria and medical conditions.
Vocal Intonation Therapy (VIT)	Controlled singing and vocal exercises to improve inflection, pitch, breath control, vocal timbre and volume. Appropriate for patients with voice disorders, medical conditions and dysarthria.

**Table 2 medicines-10-00016-t002:** Assessment of Neuropsychological clinical score.

	T0	T1
ADRS	13	10
SCNV	19	27
LCF	5	6
TOKEN TEST	3	4

Legend: T0 = baseline; T1 = follow-up after 6 months; ADRS = Aphasic Depression Rating scale; SCNV = non-verbal communication scale; LCF = Levels of Cognitive Functioning.

## Data Availability

Data sharing is not applicable to this article as no new data were created or analyzed in this study.

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
