# Peer review of "Music Therapy in Global Aphasia: A Case Report"

_medicines, 2023, doi:10.3390/medicines10020016_

Round 1

Reviewer 1 Report

The presented case report is methodologically weak, not including extensive speech and language /aphasia results and progress, memory function testing following on the longer period in the music treatment response. The presentation of results is rather weak, not including referent values for the chosen psychological tests, some MRI findings, and neurological information. What about apraxia testing? The references should be updated with recent reports on nonfluent aphasia and music treatment effects. Many questions were raised through the reading of the submitted case report. The few cases presentation would be rather more convincing, with improving background, recent reports, methodological issues, and discussion.

Author Response

Reviewer 1

The presented case report is methodologically weak, not including extensive speech and language /aphasia results and progress, memory function testing following on the longer period in the music treatment response. The presentation of results is rather weak, not including referent values for the chosen psychological tests, some MRI findings, and neurological information. What about apraxia testing? The references should be updated with recent reports on nonfluent aphasia and music treatment effects. Many questions were raised through the reading of the submitted case report. The few cases presentation would be rather more convincing, with improving background, recent reports, methodological issues, and discussion.

  • The presentation of results has been expanded and normative values for tests have been included.
    MRI findings was not included because consent for MRI was not obtained.
    Apraxia testing was not included because not in line with the aim of our study

Reviewer 2 Report

1. Title: “Music therapy in global aphasia: A case report”

2. The grammar should be revised. The authors should choose between English or American English.

The manuscript is interesting. But, it lacks a clear description of how it was performed in “musical therapy.” The authors should describe as precisely as possible. A figure would help to understand how the training was.

The reader can understand that had an improvement in the scales. But could the authors describe how this affected the individual's quality of life? What were the specific points improved with the “musical therapy.”

Could the authors provide a table with possible management choices with “musical therapy?”

What is the mechanism behind the improvement of the individual?

Author Response

Reviewer 2

  1. Title: “Music therapy in global aphasia: A case report”
  • Title has been changed

  1. The grammar should be revised. The authors should choose between English or American English.
  • Grammar has been revised

The manuscript is interesting. But, it lacks a clear description of how it was performed in “musical therapy.” The authors should describe as precisely as possible. A figure would help to understand how the training was.

  • A clearer and more in-depth description of the treatment was provided, so we did not consider necessary to include a figure

The reader can understand that had an improvement in the scales. But could the authors describe how this affected the individual's quality of life? What were the specific points improved with the “musical therapy.”

  • It was better described how the treatment improved the patient's quality of life

Could the authors provide a table with possible management choices with “musical therapy?”

  • A table was included with the different techniques of music therapy and it was explained why MUSTIM and SYCOM were chosen.

What is the mechanism behind the improvement of the individual?

  • Mechanisms underlying individual improvement have been made more explicit

Round 2

Reviewer 1 Report

It is rather unconvincing how MRI was not obtained, and no proof for the diagnosis is presented from the neurological/radiological point of view. The paper with a single case with a rehabilitative program consisting of three parts: cognitive rehabilitation, speech therapy, and neurological music therapy (NMT) could not disentangle the real effect of music therapy (also stressed in the title of the paper). 

Reviewer 2 Report

Satisfactory

Round 3

Reviewer 1 Report

The authors solved the raised issues. Thank you. The references in the text and Reference list needed to be written according to journal guidelines.